# Exercise Guidelines in Pancreatic Cancer Based on the Dietz Model

**DOI:** 10.3390/cancers17040630

**Published:** 2025-02-13

**Authors:** Philip J. Chang, Andrew E. Hendifar, Gillian Gresham, An Ngo-Huang, Paul E. Oberstein, Nathan Parker, Andrew L. Coveler

**Affiliations:** 1Cedars-Sinai Medical Center, Los Angeles, CA 90211, USA; andrew.hendifar@cshs.org (A.E.H.); gillian.gresham@cshs.org (G.G.); 2MD Anderson Cancer Center, Houston, TX 77030, USA; ango2@mdanderson.org; 3New York University, New York, NY 10012, USA; paul.oberstein@nyulangone.org; 4Moffitt Cancer Center, Tampa, FL 33612, USA; nathan.parker@moffitt.org; 5University of Washington, Seattle, WA 98105, USA; acoveler@uw.edu

**Keywords:** gastrointestinal cancer, pancreatic cancer, exercise guidelines, exercise

## Abstract

Pancreatic and gastrointestinal cancers commonly occur in the elderly and in patients with lower levels of physical function. Physical activity and exercise are one of the primary interventions in combating frailty and disability; however, guidelines are lacking in this patient population. Here, we propose practical exercise guidelines for every phase of disease in these cancers based on the Dietz model. During the prehabilitation phase, patients may optimize their physical abilities before undergoing surgery. Restorative rehabilitation often given through skilled therapies can help patients to recover any function lost from chemotherapy, radiation therapy, and surgery. Supportive rehabilitation can allow patients to continue with regular physical activity during stable or ongoing disease and palliative rehabilitation can help patients maintain the highest possible quality of life when they enter the end stage of life. No matter the stage of disease or phase of treatment, physical activity and exercise have important roles in maintaining well-being.

## 1. Introduction

Despite advances in therapy, the mortality associated with gastrointestinal cancers, particularly pancreatic ductal adenocarcinoma (PDAC), continues to be substantial. PDAC is currently projected to be the second leading cause of cancer deaths by 2040 [1], and colorectal cancers have shown increasing incidence of early onset [2]. These malignancies are associated with a disproportionate rate of weight loss, pain, and debility [3]. In addition, chemotherapy, comprised of combinatorial cytotoxic regimens, is the mainstay of therapy and associated with significant toxicities [4].

Weight loss, sarcopenia, and frailty are highly prevalent among patients with gastrointestinal cancers [5]. Up to 80% of PDAC patients have significant weight loss on diagnosis and meet criteria for cancer cachexia [6]. Frailty syndrome, a state of extreme vulnerability to stressors that leads to adverse health outcomes, is also highly prevalent in this population. These features are associated with significant declines in physical activity and performance, therapeutic toxicity, and perioperative complications [3,4,7,8,9]. They also correlate with shorter progression-free survival (PFS), shorter overall survival (OS), and lower quality of life (QOL) [10].

Physical activity has been associated with decreased cancer incidence and improved outcomes, specifically concerning quality of life [11,12,13,14,15]. In addition, preclinical studies have shown that aerobic exercise reduced PDAC and colon tumor growth by modulating systemic and intra-tumoral immunity [3,16,17]. However, recommendations for the incorporation of physical activity into the care of these cancers to combat weight loss, frailty, and sarcopenia are lacking [18]. Here, we briefly discuss the benefits of exercise in gastrointestinal cancers followed by a practical approach to physical activity based on the Dietz model of rehabilitation in cancer.

## 2. Review of Preclinical and Clinical Exercise Benefits

Preclinical and some clinical evidence suggests that exercise is useful as an adjuvant to chemotherapy by improving chemotherapy delivery and anti-tumor efficacy. Tumor vasculature is highly disorganized, with a dense stromal component that compresses vasculature, resulting in disrupted blood flow that impedes drug delivery. Moderate-intensity aerobic exercise has been shown to induce vascular normalization in mouse models and humans [19].

One mechanism by which exercise mediates vascular normalization is by shear stress, the mechanical stimuli exerted on endothelial cells by increased blood flow. In a mouse model utilizing moderate-intensity treadmill running to increase blood flow in mice with PDAC tumors, tumor growth was significantly inhibited in mice treated with chemotherapy and aerobic exercise compared to mice treated with chemotherapy alone [20]. In another study, mice bearing patient-derived xenograft tumors were treated with moderate-intensity treadmill exercise, gemcitabine, a combination of exercise plus gemcitabine, or phosphate-buffered saline [21]. Exercise alone had no effect on tumor growth; however, exercise plus gemcitabine caused faster tumor regression and delay in tumor regrowth. In a human study with 70 participants undergoing neoadjuvant therapy for resectable PDAC, patients were prescribed at least 120 min of moderate-intensity, home-based exercise (60 min of aerobic and 60 min of strengthening) per week [21]. Of the 70 patients, 33 underwent resection and 23 had tissue and activity data. Of these 23 patients, the mean time of exercise was 15 weeks (±6.5), with an average of 170 ± 80 min per week of exercise. There was no change in TNM stage or regression, despite seeing an increase in micro-vessel density [21].

In addition to vascular changes, there is a growing awareness that exercise may modulate tumor growth by impacting the immune microenvironment [22,23]. Kurz and colleagues employed a low-intensity treadmill running exercise regimen in several mouse models and demonstrated that this intervention delayed the onset of pancreatic cancer and enhanced the impact of chemotherapy and immunotherapy in controlling tumors [16]. They identified alterations in anti-tumor immunity that were mediated by interleukin-15 signaling [16]. Exercise has also been shown to control tumor growth through the mobilization of natural killer cells [24].

In the small, randomized SUPPORT study, patients assigned to a supervised resistance training program had improved muscle strength in some muscle groups compared to those in a home-based resistance training or control (no training) group [25]. Interestingly, those in the supervised resistance group also had alterations in the tryptophan/kynurenine pathway that may have been associated with improved immune response to tumors [26].

## 3. Exercise Guidelines

The Dietz framework for rehabilitation in cancer was first described in the 1980s and describes four phases of rehabilitation including preventative rehabilitation, restorative rehabilitation, supportive rehabilitation, and palliative rehabilitation (Box 1). We describe physical activity guidelines in the context of these phases, with a focus on the specific challenges of gastrointestinal cancers. Table 1 highlights relevant clinical trials in pancreatic cancer demonstrating the benefits of exercise programming.

Box 1Key exercise pointsPrehabilitation
Provide education on prehab benefitsGoal of 150 min of moderate-intensity aerobic exercise per weekGoal of 2–3 days per week of resistance trainingRestorative Rehab
Encourage early postoperative mobilizationUtilize PT (inpatient, home, or outpatient) for gait difficulties and decon-ditioningUtilize OT (inpatient, home, or outpatient) for ADL deficitsSupportive Rehab
Assess for functional deficits at regular intervalsRoutinely discuss fine motor function, gait, balance, and falls, which may occur secondary to chemotherapy neuropathy**End of Life**
Focus should transition to comfort while trying to maximize functional independence-Consider use of home PT/OT for caregiver training

**Table 1 cancers-17-00630-t001:** EX (exercise); CON (control).

Study	Design	Sample Size	Intervention	Outcome Measures	Results
Florez et al., 2019 [21]	Prospective single arm with historical control	23 EX, 13 historical CON	Home-based: 60 min aerobic; 60 min strength per week	Tumor vascularity	Twice as many vessels per field in EX group on pathology
Ausania et al., 2019 [15]	RCT	18 EX, 22 CON	Nutritional support: 60 min supervised exercise × 5 days then unsupervised home exercises	Post-op complication rate (Dindo–Clavien classification)	Reduction in delayed gastric emptying in EX group
Nakajima et al., 2019 [27]	Prospective single arm with historical control	76 EX, 142 historical CON	Home-based unsupervised: 60 min aerobic and resistance 3× per week	Length of hospital stay	Median 23 days vs. 30 days in EX vs. CON groups, respectively
Ngo-Huang et al., 2023 [28]	2-arm RCT	76 EX, 76 CON	Home-based: ≥30 min aerobic 5× per week; ≥2× per week resistance exercise sessions	6 min walking distance	Mean change 28.3 ± 68.2 m in EX vs. 17.8 ± 56.7 m in CON
Okada et al., 2022 [29]	Prospective single arm	43 EX	Inpatient immediate post-operative training with aerobic and resistance exercise	Completion rate of S-1 therapy	93% completion rate, which exceeded the threshold rate of 53%
Streckmann et al., 2024 [30]	3-arm RCT	55 EX, 53 VT, 50 CON	Sensorimotor Training	Development of chemotherapy neuropathy	Decreased incidence of neuropathy in EX and VT groups

## 4. Prehabilitation (Preventative Rehab)

Pretreatment exercise training, or exercise prehabilitation, is an increasingly adopted strategy to improve the physiologic status of individuals with cancer. Given the importance of maintaining aerobic and muscular fitness and physical functioning throughout cancer survivorship, prehabilitation can be applied as patients approach various therapeutic contexts including chemotherapy, radiation therapy, immunotherapy, targeted therapy, and surgery.

The concept of prehabilitation in medicine emanates primarily from surgical literature. Surgery can have profound and deleterious effects on quality of life through unintended consequences including increased fatigue, reduced physical activity and energy intake, altered sleep, and impaired physical functioning [31]. Suboptimal fitness has been associated with an increased risk of surgical complications, hospital readmissions, nutritional challenges, and requirements for intensive care that can prolong post-operative bed rest and slow recovery [31].

To date, the preoperative context has been the most frequent target for exercise intervention in pancreatic cancer, including interventions delivered concurrently with neoadjuvant therapy. As with other cancer diagnoses, preoperative exercise training capitalizes on a convenient and important window for patients with gastrointestinal cancers. A recent systematic review reported median wait times between workup and surgical resection ranging from 14 to 42 days among patients undergoing upfront operations for resectable pancreatic cancer [32]. These lead times generally align with durations of cancer prehabilitation programs with favorable outcomes involving postoperative functional recovery [33]. Exercise training concurrent with neoadjuvant therapies may be particularly important to improve treatment tolerance and mitigate potential declines in physical functioning and fitness prior to surgery. A new cancer diagnosis represents a major life event and patients may be particularly motivated to make lifestyle changes (e.g., increasing physical activity) that can improve their trajectory and help them maintain an improved quality of life.

An increasing evidence base demonstrates the feasibility, safety, and potential benefits of gastrointestinal cancer prehabilitation. In a randomized trial of a hybrid (in-person supervised and home-based unsupervised) exercise program including both aerobic and resistance training, Ausania et al. (2019) reported a lower rate of delayed gastric emptying in prehabilitation participants compared to those receiving usual care, but no other significant differences in other complications or length of postoperative stay [15]. Nakajima and colleagues (2019) demonstrated improvements in 6 min walking distance and body composition (muscle-to-fat ratio) among participants in a home-based, unsupervised aerobic and resistance training program prior to major abdominal surgeries for hepato-pancreato-biliary malignancies [27]. Comparing outcomes to a propensity score-matched historical group, the study found reduced length of postoperative stay but no difference in surgical complications [27]. A multicenter, randomized, controlled trial found that participants in a multimodal prehabilitation program experienced fewer severe complications postoperatively compared to standard care in patients undergoing colorectal surgery [34]. In a prospective, single-arm study of home-based aerobic and resistance training concurrent with neoadjuvant chemotherapy and chemoradiation therapy, Parker et al. (2019) demonstrated strong aerobic exercise adherence and favorable outcomes involving physical functioning, health-related quality of life, and tumor vascularity [35]. In comparison to a retrospectively identified, non-prehabilitation group, Parker et al. (2020) also found the mitigation of skeletal muscle loss among prehabilitation participants [36]. This group followed the single-arm prehabilitation trial with a randomized trial comparing outcomes among participants in a home-based, unsupervised aerobic and resistance training program to those among patients encouraged to engage in aerobic exercise. Prehabilitation participants engaged in more weekly strength training sessions than enhanced usual care participants, but the two groups engaged in similar levels of Fitbit-measured physical activity. Both groups demonstrated significant improvements in aerobic fitness, as measured by 6 min walk tests. In pooled analyses, higher levels of self-reported exercise and Fitbit-measured physical activity were predictive of better outcomes including physical functioning and fitness [28]. Of particular note, a recent study demonstrated that sensorimotor training consisting of progressive balance exercises during chemotherapy with vinca alkaloids or oxaliplatin reduced the incidence of chemotherapy neuropathy to 30% compared to 70% in the control group [30].

Despite its documented benefits, exercise training for prehabilitation is not yet included in routine clinical practice. There exist no consensus guidelines regarding exercise modalities or program delivery strategies in cancer prehabilitation [31,33], but programs tend to follow the American College of Sports Medicine (ACSM) Exercise Guidelines for Cancer Survivors and incorporate both aerobic training and resistance training [25].

## 5. Restorative Rehabilitation

Restorative rehabilitation occurs after a decline in function secondary to disease or treatment. There are similar benefits for physical activity postoperatively, during adjuvant therapy, and in survivorship for patients with gastrointestinal cancers. Postoperatively, progressive mobilization on the same day of surgery compared to the day after surgery showed benefits in oxygenation [37]. Postoperative activity has been shown to improve recovery of gastrointestinal function, length of stay, functional recovery, pain [38], risk of readmission, and health-related quality of life. After surgery, mobilization should be initiated as soon as is safe. Ambulation can be accomplished by having nurses, nursing aides, and/or therapy technicians supervise the ambulation if no significant functional concerns are noted. Strategies to enhance postoperative mobility may include posting ambulation goals (via signs and/or activity log) in the patient’s room, providing portable cyclers in the room, and, if there is significant postoperative abdominal pain, providing an abdominal binder (if not contraindicated). One study showed that the use of activity trackers after laparoscopic surgery resulted in increased step counts and activity minutes [39]. If there is concern about gait instability and fall risk, an early referral for acute care physical therapy (PT) should be initiated. If the patient has difficulty with activities of daily living (ADL), occupational therapy (OT) should be consulted to evaluate the patient’s ability to complete ADL (feeding, grooming, upper and lower body dressing, transfers for self-care, toileting, bathing). OT trains patients and caregivers for mobility-related ADL, provides breathing and energy conservation recommendations, and may provide adaptive equipment (grabber, long-handed sponges, sock aides). Both OT and PT may assess durable medical equipment needs (wheelchair, rollator, standard walker, cane, bedside commode, shower chair, hospital bed) to support a safe discharge to home.

Once patients are medically stable for discharge, if mobility concerns persist, additional rehabilitation options may include prescriptions for outpatient PT/OT and home health services (including home PT/OT) or referrals to an inpatient facility for rehabilitation (skilled nursing facility, acute inpatient rehabilitation, and/or long-term acute care hospital). Improving postoperative mobilization, including physical activity recommendations after discharge from the hospital leading to starting adjuvant chemotherapy, is essential for functional recovery. Supervised exercise during adjuvant chemotherapy resulted in improved rates of S-1 adjuvant chemotherapy completion and reduced frailty [29]. When supervised exercise programs are not readily accessible, patients are encouraged to ambulate at home multiple times per day for the goal of 30 min of mobility per day for six to eight weeks after surgery and refrain from any lifting greater than 10 pounds. A systematic review of exercise for patients with PDAC found that during adjuvant treatment, exercise is safe and effective in mitigating impaired physical function, quality of life, and fatigue [40].

## 6. Supportive Rehabilitation

Supportive rehabilitation occurs during stable residual disease or remission. This scenario is frequent in advanced gastrointestinal cancers, where patients may be on indefinite treatment. In this survivorship stage, the primary goals of physical activity should be to optimize quality of life and re-integrate into recreational and valued roles.

This stage may not be distinct from the restorative phase and continual reassessment is paramount. During supportive rehabilitation, declines in functional status may occur with worsening liver function, uncontrolled pain, worsening chemotherapy side effects, neuropathy, infections, and prolonged hospitalizations. To assess for such declines, assessment of balance, proprioception, gait, muscle strength, and sensation should occur at regular intervals.

Chemotherapy-induced peripheral neuropathy (CIPN) is a common side effect of oxaliplatin (a component of FOLFIRINOX). In patients with PDAC receiving FOLFIRINOX, 61% had any-grade CIPN and 9% had high-grade CIPN [41,42]. Symptoms include numbness, paresthesia, dysesthesias, or loss of sensation in the distal extremities. CIPN can lead to the loss of balance and falls so early diagnosis and intervention is vital to minimize complications. There is limited strong evidence for CIPN treatments. Patients with peripheral CIPN symptoms are encouraged to wear cotton socks and gloves and avoid extreme heat or cold temperatures. For patients where CIPN has affected their gait and balance, physical therapy and occupational therapy interventions may include gait training; fall recovery training; modalities such as Kinesio taping of the hands, feet, and ankles; hand therapy to preserve or restore manual dexterity; sensory feedback treatments; and transcutaneous electrical neurostimulation (TENS) treatments to treat pain. Other physical modalities to consider include acupuncture and massage therapy. Pharmacologic treatment for pain/discomfort may include anti-neuropathic agents, with the strongest evidence for duloxetine [43]. There is also emerging evidence for cryotherapy and acupuncture. In one trial, patients receiving oxaliplatin for malignancies of the GI system were randomized to continuous cooling of the hands and feet during their chemotherapy infusion versus usual care (no cooling). Patients in the intervention group had less numbness or tingling in the fingers and toes, pain, and cold sensitivity [44]. Currently pending or in-progress clinical trials include the use of riluzole to prevent CIPN [45]; a multi-site, randomized trial of donepezil to treat oxaliplatin-induced CIPN [46]; and the use of acupuncture with acupressure along with cryotherapy [47].

To maintain mobility and quality of life during treatment, patients should be encouraged to meet ACSM exercise guidelines. Any level of tolerable physical activity, even if small, should be encouraged. Equally important is having patients try to return to their valued roles and identifying gaps where patients functionally are versus where they want to be. Occupational therapists are an indispensable resource in this area and there should be a low threshold for referral. While the effects of exercise on outcomes for long-term pancreatic cancer survivors is currently limited, multiple studies have shown associations with increased survival and increased levels of physical activity [48,49].

## 7. Hospice and the End of Life

Although palliative care should not be confused with hospice and end-of-life care, in the context of the Dietz framework, palliative rehabilitation applies to those at the end stage of life. For patients who no longer wish to pursue disease-modifying treatment but want to improve their performance status, the guidelines for restorative and supportive rehabilitation apply. However, for patients who are significantly debilitated with poorly controlled symptoms, the goal of rehabilitation should focus on comfort.

Closer to the end of life, functional impairments may become more global, encompassing difficulty with transfers, sitting, standing, ambulation, and the completion of ADL [50,51,52]. These deficits may further be compounded by side effects from treatment including peripheral neuropathy, cancer-related pain, and sedation from opioid therapy. Despite these impairments, studies have shown that patients in hospice settings and palliative care units may improve functional performance with ADL and transfers [53,54,55]. While maximizing functional independence is important, the primary goals should be to avoid falls, alleviate symptoms, and reduce caregiver burden.

One of the primary ways these goals can be achieved is through caregiver training to properly and safely assist patients with transfers, ambulation, and ADL, which can be completed through inpatient acute care or inpatient rehabilitation, at a skilled nursing facility, or in the home setting. A case study demonstrated that proper training improves caregiver confidence in assisting with transfers [56], and a short, palliative inpatient rehabilitation program for deconditioning led to a safe home discharge with caregiver assistance [57]. While some hospices may not offer skilled rehabilitation, many may allot 1–2 sessions for caregiver functional training. Additionally, if skilled therapists are available in hospices, physical modalities such as electrical stimulation and compression bandaging may provide symptom relief [58,59]. Patients and caregivers should be educated on the availability of these services so that they can advocate for themselves. They should also be educated that the goal is not for functional recovery but for symptomatic relief and caregiver training. For patients already receiving home hospice care, discussion with the hospice social worker, nurse, or physician on the benefits of a skilled therapy referral may be helpful.

## 8. Role of Wearable Technology

The majority of cancer patients do not meet exercise guidelines and adherence to activity remains low [60,61]. Furthermore, it is difficult to obtain accurate estimates of a patient’s daily activity as patients can experience dynamic changes in their activity patterns throughout their cancer experience. The use of wearable activity monitors can play an important role in motivating physical activity while allowing for the passive collection of objective, continuous measures of physical activity (physical activity duration and intensity, time spent sedentary, steps per day, sleep, heart rate, and other metrics), providing a more complete picture of the patient’s activity in their free-living environment [62].

There is growing interest in including daily stepping recommendations in exercise guidelines as step counts are easily interpretable and widely available through wearables and other activity-tracking devices [63]. However, challenges including the lack of standardization across devices and metrics and variability across populations have made it difficult to identify and recommend specific activity targets and stepping dosages to cancer survivors. As more research emerges supporting the role of activity monitoring in promoting physical activity, more data informing guideline recommendations will also become available [63].

Studies that have used wearables and report on daily activity metrics have focused on postoperative outcomes in patients undergoing surgery [28,64,65,66,67], while few have evaluated the use of wearables in advanced cancer patients and those undergoing active treatment [62,68]. These studies have reported a range of approximately 3700–6000 steps per day, depending on disease stage and treatment, although significant heterogeneity exists across populations (e.g., stage and treatment), device types, and wear time. To date, no studies have provided tailored recommendations or daily activity targets based on wearable activity monitors for pancreatic cancer patients. While future guidelines should consider the incorporation of wearable activity metrics to provide activity targets, tracking step counts can be helpful in allowing patients to assess their progress (Box 2).

Box 2Tips for exercise counselingTips
Avoid reciting exercise guidelines and create customized programs based on individual needsStart with small, achievable goals (i.e., walking 5 min twice a day) and gradually increase themUtilize existing web-based or Electronic Medical Record (EMR)-embedded tools when creating resistance programs for patientsTrack step counts with smartphones and wearable technology to assess progress

## 9. Conclusions

Patients with gastrointestinal cancers become debilitated secondary to chemotherapy-induced fatigue and muscle loss and due to post-surgical deconditioning and primary cancer effects. Here, we have outlined practical exercise goals and interventions at every phase of illness (Table 2). Exercise training concurrent with neoadjuvant therapies is particularly important to improve treatment tolerance and mitigate potential declines in physical functioning and fitness prior to surgery. Prehabilitation has demonstrated lower rates of delayed gastric emptying, the mitigation of skeletal muscle loss, and reduced lengths of postoperative stay. In postoperative and/or post-adjuvant therapy, there is time for restorative and supportive rehabilitation. We should not assume that patients will automatically exercise and recover to their baseline functional levels without assistance. This time is opportune to build strength back regardless if patients remain cancer-free or relapse. Finally, during and after therapy, assessments of balance, proprioception, gait, muscle strength, and sensation should occur at regular intervals and ideally by a trained professional.

## Figures and Tables

**Table 2 cancers-17-00630-t002:** ADL (activities of daily living); ECOG (Eastern Cooperative Oncology Group Performance Status); QOL (quality of life); ACSM (American College of Sports Medicine); SMT (sensorimotor training); VT (vibration training).

	Aims/Goals	Screening	Monitoring	Education	Intervention	Issues
**Prehabilitation**	Optimize functional status prior to surgery	Examine gait, balance, independence with ADL, ECOG score	Continue to monitor gait, ADL, balance, ADL function, and ECOG during routine follow-ups	Discuss with patients the risk of falls and injury; advise to “start low and go slow”; discuss QOL benefits of exercise	If ECOG 0/1 trial, home-based program per ACSM 2019 guidelines; if ECOG 1-3, may refer to home or outpatient physical therapy	May not be feasible in patients planned for upfront surgery depending on lead times; consider implementing SMT or VT to prevent neuropathy during neoadjuvant treatment
**Restorative Rehab**	Identify and rehabilitate functional deficits occurring during treatment	Examine proximal/distal muscle strength, fine motor coordination, gait, and balance (tandem stance and single leg)	Continue to monitor strength, fine motor coordination, gait, and balance and conduct routine follow-ups	Discuss with patients that the effects of chemotherapy may be cumulative so continued exercise is essential	If deficits in strength, gait, or balance are detected, then refer to skilled therapy; if no deficits, may continue on home program—recommend ambulatory assistive devices when applicable	Start to monitor for chemotherapy-induced peripheral neuropathy (balance, proprioception, fine motor skills); consider implementing SMT or VT to prevent neuropathy during adjuvant treatment
**Supportive Rehab**	Maintain functional status during ongoing treatments	Examine proximal/distal muscle strength, fine motor coordination, gait, and balance (tandem stance and single leg)	Continue to monitor strength, fine motor coordination, gait, and balance and conduct routine follow-ups	Encourage the maintenance of a physically active lifestyle within functional limits	Continued home-based programming vs. continued physical therapy or institutional or community programs when available—recommend ambulatory assistive devices when applicable	Continue to monitor for chemotherapy-induced peripheral neuropathy
**End of Life**	Transition focus of care to comfort while maintaining physical function within goals of care	Examine gait, sit-to-stand transfers, and ECOG	Continue to monitor gait, sit-to-stand transfers, and ECOG at routine follow-ups	Discuss with patients the transition of goals and what their goals for care are; discuss a change of goal to comfort as opposed to functional independence	Home-based or outpatient skilled therapies; consider need for durable medical equipment including hospital bed, assistive device, shower chair, raised toilet seat, and grab bars	Consider transition to hospice care where skilled therapy services may still be available

## Data Availability

Not applicable.

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
