# Peer review of "Exercise Guidelines in Pancreatic Cancer Based on the Dietz Model"

_cancers, 2025, doi:10.3390/cancers17040630_

Round 1
Reviewer 1 Report
Comments and Suggestions for Authors
The paper "Exercise Guidelines in Pancreatic Cancer Based on the Dietz Model" by Chang et al. presents a perspective on integrating exercise into the care of pancreatic cancer patients, structured around the four rehabilitation phases of the Dietz model: preventative, restorative, supportive, and palliative.
Major points:
The authors utilize clinical and preclinical studies to highlight the benefits of exercise, such as improved chemotherapy delivery, enhanced immune response, and better quality of life.
Specific exercise types, durations, and intensities are proposed, providing actionable recommendations.
However, including a table summarizing current clinical trials related to exercise in pancreatic cancer would provide valuable context.
A discussion on long-term pancreatic cancer survivors and the relevance of exercise for their outcomes should also be included.
Finally, the visual layout of the two boxes summarizing key points could be refined to make them more appealing and engaging for readers.
Minor Points:
Page 3, line 140: Correct “patinets” to “patients.”
Page 5, line 228: Revise “Pharmacologic treatment for pain/discomfort may include neuropathic agents with the strongest evidence for duloxetine(42)” to “Pharmacologic treatment for pain/discomfort may include anti-neuropathic agents, with the strongest evidence for duloxetine.”
Page 7: Write out "EMR" as “Electronic Medical Records” when first mentioned.
Page 7: Revise “Avoid reciting exercise guidelines and create customized programs based off individual needs” to “Avoid reciting exercise guidelines and create customized programs based on individual needs.”
Recommendation: Accept with minor revisions.
Author Response
Reviewer 1:
Comments and Suggestions for Authors
The paper "Exercise Guidelines in Pancreatic Cancer Based on the Dietz Model" by Chang et al. presents a perspective on integrating exercise into the care of pancreatic cancer patients, structured around the four rehabilitation phases of the Dietz model: preventative, restorative, supportive, and palliative.
Major points:
The authors utilize clinical and preclinical studies to highlight the benefits of exercise, such as improved chemotherapy delivery, enhanced immune response, and better quality of life.
Specific exercise types, durations, and intensities are proposed, providing actionable recommendations.
However, including a table summarizing current clinical trials related to exercise in pancreatic cancer would provide valuable context.
-A table summarizing recent clinical trials has been added following Section 3
A discussion on long-term pancreatic cancer survivors and the relevance of exercise for their outcomes should also be included.
-A statement with two additional references have been added to the end of the “Supportive Rehabilitation” section.
Finally, the visual layout of the two boxes summarizing key points could be refined to make them more appealing and engaging for readers.
-We have bolded the headings, added bullets for each point and underlined key phrases
Minor Points:
Page 3, line 140: Correct “patinets” to “patients.”
-Corrected
Page 5, line 228: Revise “Pharmacologic treatment for pain/discomfort may include neuropathic agents with the strongest evidence for duloxetine(42)” to “Pharmacologic treatment for pain/discomfort may include anti-neuropathic agents, with the strongest evidence for duloxetine.”
-This has been rephrased as recommended
Page 7: Write out "EMR" as “Electronic Medical Records” when first mentioned.
-This has been written out as recommended
Page 7: Revise “Avoid reciting exercise guidelines and create customized programs based off individual needs” to “Avoid reciting exercise guidelines and create customized programs based on individual needs.”
-This has been revised as recommended
Recommendation: Accept with minor revisions.
Reviewer 2 Report
Comments and Suggestions for Authors
The review "Exercise Guidelines in Pancreatic Cancer Based on the Dietz Model" describes the 4 phases of rehab during cancer therapy. The text is supported by the evidence from various studies that exercise helps. The review is concise, however, including the clinical aspects will be interesting. Please include how to screen, monitor, educate, intervene, prevent the effects of chemotherapy, avoid functional impairment via intervention, and maintain function and quality of life during and in these 4 phases of rehab care. Please also include what should be the aims of these 4 phases of rehab, the limitations, issues, and the outcome. Please also include strategies or step to be taken to improve the clinical outcomes.
Author Response
Reviewer 2
The review "Exercise Guidelines in Pancreatic Cancer Based on the Dietz Model" describes the 4 phases of rehab during cancer therapy. The text is supported by the evidence from various studies that exercise helps. The review is concise, however, including the clinical aspects will be interesting.
Please include how to screen, monitor, educate, intervene, prevent the effects of chemotherapy, avoid functional impairment via intervention, and maintain function and quality of life during and in these 4 phases of rehab care.
-We are grateful for this excellent recommendation. Table 2 has been added to address these points.
Please also include what should be the aims of these 4 phases of rehab, the limitations, issues, and the outcome.
-Table 2 has been added to address these points
Please also include strategies or step to be taken to improve the clinical outcomes.
-We feel that this is addressed in Table 2, Box 1 and Box 2
Reviewer 3 Report
Comments and Suggestions for Authors
This study was aimed to investigate current status of rehabilitation especially in patients with pancreatic cancer. The manuscript was written by narrative review style. There have been not so many reports to be investigated and reported at the present time in this field of rehabilitation role.
This manuscript was well-written by authors Therefore, this perspective might be very useful in the field of clinical implication of rehabilitation.
Author Response
Reviewer 3
This study was aimed to investigate current status of rehabilitation especially in patients with pancreatic cancer. The manuscript was written by narrative review style. There have been not so many reports to be investigated and reported at the present time in this field of rehabilitation role.
This manuscript was well-written by authors Therefore, this perspective might be very useful in the field of clinical implication of rehabilitation.
-Thank you for your generous feedback!
Round 2
Reviewer 2 Report
Comments and Suggestions for Authors
Please mention Box 1 and Table 2 in the text as per their relevance of the text.
Author Response
Please mention Box 1 and Table 2 in the text as per their relevance of the text.
-Box 1 and Table 2 are now mentioned in the text